# SUPER FLOATING-POINT (SuFP): EFFICIENT TO ALL. MULTI-REGION PIECEWISE QUANTIZATION USING SCALABLE BIAS WITH HARDWARE OPTIMIZATION

## ABSTRACT

As Deep Neural Networks (DNNs) revolutionize various application domains, their model size and computational demand also increase exponentially. In response to these challenges, various quantization techniques have emerged as highly effective solutions. However, quantization methods using conventional data types, including integer or floating-point, face certain limitations in balancing between accuracy drop and computational benefit. In light of the advent of hardware accelerator design for AI processing, quantization research has entered a new phase: custom data types and specialized hardware have emerged as innovative alternatives. Particularly, piecewise quantization and block floating-point quantization exhibit notable performance and efficiency improvements, but they still suffer from handling outliers with huge dynamic ranges. To solve this issue, we introduce Super Floating-Point (SuFP), a breakthrough data type and quantization method that improves both memory footprint and logic efficiency without compromising model accuracy. The key idea of SuFP is multi-region piecewise quantization using a tensor-wise scalable bias. It can configure an optimized precision for each region to capture both dense near-zero data and outliers. In addition, the scalable bias offers flexible adaptability to diverse data distributions, requiring only a single addition operation at the tensor level. Furthermore, the tailored hardware for SuFP employs only integer arithmetic units and shifters, facilitating a highly compact hardware realization. Our experimental results show that SuFP quantization achieves accuracy performance on par with, and in some cases even exceeds, that of full precision floating-point (FP32) across vision, language, and generative model benchmarks. Its computational capability and energy efficiency have shown improvements, with a $9.00\times$ and $17.04\times$ enhancement over FP32 implementations. These improvements are notable when compared to state-of-the-art MSFP and BSFP, which show up to $7.20\times$ and up to $8.27\times$, respectively.

## 1 INTRODUCTION

Deep Neural Networks (DNNs) have demonstrated exceptional performance across a wide range of applications, including image classification (Deng et al., 2009) and natural language processing (Chowdhary & Chowdhary, 2020). Moreover, DNNs are now excelling in state-of-the-art generative models, such as text-to-image models (Rombach et al., 2022) and Large Language Models (LLMs) (Touvron et al., 2023). This extended applicability further elevates the standing and importance of DNNs within the broader AI landscape.

However, the exponential increase in model size and computational complexity results in a bigger memory footprint and computational capacity requirements (Gholami et al., 2021b). This growth creates significant bottlenecks in DNN inference tasks, leading to longer delays and slower response time on every computational hardware from server to edge devices. In response to these challenges, quantization has been recognized as an effective approach that delivers significant performance improvements.

Ongoing research is actively exploring various approaches for the quantization methods. A key challenge is balancing accuracy loss from low-bit operations with improved memory footprint and com-

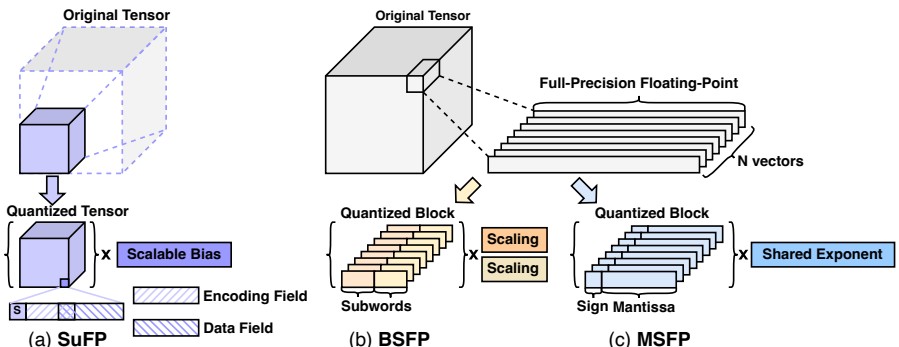

Figure 1: Comparison of the quantization processes between (a) the proposed Super Floating-Point, (b) Block and Subword-scaling Floating-Point, and (c) Microsoft Floating-Point.

putational efficiency. One possible approach to address this challenge is quantization-aware training (QAT). However, QAT can only be applied to models with publicly disclosed training datasets and parameters, significantly limiting its scope of use. Due to such constraints, many users tend to prefer post-training quantization (PTQ), which directly applies to pre-trained weights and activations. Although PTQ is a straightforward method, it is likely to result in a more significant accuracy loss than QAT. In light of this, extensive quantization research has been conducted for conventional data types including integer and floating-point. However, integer quantization, for instance, (Krishnamoorthi, 2018; Nagel et al., 2021; Gholami et al., 2021a), inherently possesses limitations in terms of accuracy due to its limited dynamic range and challenges in implementing non-uniform quantization. On the other hand, although floating-point quantization, including (Wang et al., 2018; Sun et al., 2019), can mitigate some of these issues, insufficient bit-width of mantissa in floating-point quantization limits its application.

In addition, custom data types are emerging along with the development of novel processing units for AI workloads, such as (Jain et al., 2019; Guo et al., 2022; 2023), to overcome the limitations of conventional data types. Custom data type can be fitted to model parameters and activation data distribution, minimizing performance degradation. Also, custom hardware guarantees hardware efficiency by executing operations tailored to these custom data types (Park et al., 2018; Zadeh et al., 2020).

Among various approaches, piecewise quantization and block floating-point quantization stand out for their performance and hardware efficiency. Piecewise quantization (Fang et al., 2020; Jain et al., 2019; Yuan et al., 2022) subdivides a dynamic range into several regions to capture the entire data distribution. On the other hand, block floating-point quantization (Darvish Rouhani et al., 2020; Lo et al., 2022) utilizes small-sized blocks and a shared scaling factor, making it applicable to various data distributions. Nonetheless, both piecewise and block floating-point methods struggle to ensure optimal precision in a whole range of data, including near-zero values and outliers, which is crucial for attaining the overall model accuracy. Specifically, piecewise quantization methods should consider the trade-off between the number of regions and the bit-width allocated to each region within a constrained bit budget. The block floating-point exhibits a uniform distribution within a block that shares a common scaling factor. This characteristic makes it challenging to cover various dynamic ranges. Furthermore, reducing the block size to address this issue increases memory footprint overhead.

In this paper, we introduce Super Floating-Point (SuFP), a breakthrough data type that is "efficient to all" in terms of accuracy, adaptability, and hardware. SuFP utilizes a variable encoding field that facilitates multi-region piecewise quantization, allowing for the effective representation of the entire data distribution. Specifically, different precisions are set to capture the dense region in the near-zero range and the sparse region containing outliers, which maximize the numerical representation efficiency within a limited number of bit-width. Figure 1 (a) highlights the key concept behind SuFP: the combination of a variable encoding field and a variable data field with a tensor-wise scalable bias. In contrast to the block floating-point approach, which applies a scaling factor on a block-by-block basis (Figure 1 (b) and (c)), SuFP employs a scalable bias across the tensor level.

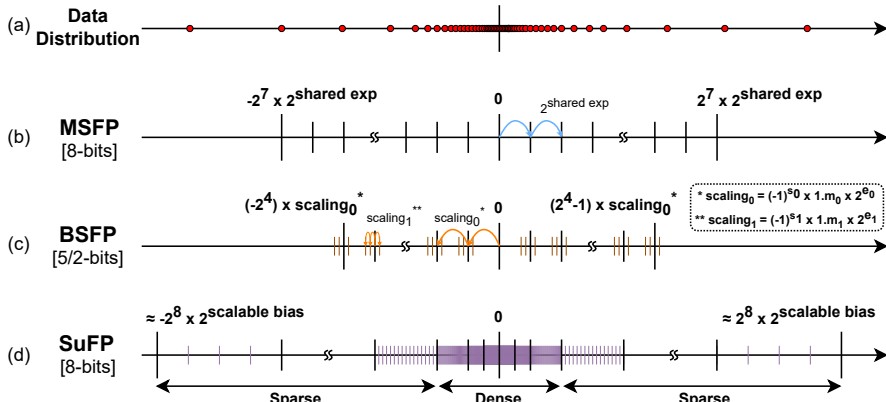

Figure 2: Data distribution comparison between (a) real data distribution with (b) 8-bit MSFP, (c) 7-bit BSFP and our (d) 8-bit SuFP.

Figure 2 illustrates the benefits of using scalable bias and multi-region method, with SuFP optimally providing granularity in each region. In this way, SuFP results in low quantization errors over diverse data distributions, implying its broad applicability. Moreover, SuFP is highly hardware-efficient; the tensor-wise scalable bias requires only a single integer addition in a tensor-level operation, enabling coverage of a diverse dynamic range with a negligible hardware overhead. Additionally, the tailored hardware for SuFP consists solely of an integer arithmetic unit and a shifter, allowing for a highly compact hardware configuration.

In summary, the contributions of SuFP are:

- Efficient To Accuracy: SuFP utilizes multi-region piecewise floating-point quantization based on the variable encoding field to ensure adequate data granularity in both dense and sparse data distribution regions. This approach minimizes quantization error by optimizing the granularity of data distribution for each stage.

- Efficient To Adaptability: SuFP adopts a tensor-wise scalable bias, enabling precise adjustment to its range in alignment with the dynamic range of any data distribution. By setting the bias appropriately, SuFP can accurately represent various data distributions across diverse DNN models. This approach highlights the enhanced adaptability of SuFP.

- Efficient To Hardware: SuFP excels in hardware efficiency, demonstrating superior computational performance. The custom Processing Element (PE) designed for SuFP computations achieves optimal computational efficiency, affirming that SuFP not only assures high accuracy but also provides a hardware-friendly data type.

## 2    RELATED WORKS

With the exponential growth in model size, various efforts have been made within the deep neural network community to optimize performance. Notably, quantization research attempts to balance reducing memory footprint and preserving model accuracy. In this section, we explore three primary quantization methodologies: low-bit quantization, piecewise quantization, and block floating point quantization.

Low-bit quantization techniques for neural networks can broadly be categorized into two primary methodologies: integer (INT) quantization and floating-point quantization. Numerous works have delved into these methodologies, with references such as (Xiao et al., 2023; Wu et al., 2020; Yao et al., 2022; Dettmers et al., 2022; Frantar et al., 2022) focusing on INT quantization, and (Kuzmin et al., 2022; Zhang et al., 2023; Wu et al., 2023; Micikevicius et al., 2022; Sun et al., 2019) on floating-point quantization strategies.

INT quantization has simple operations and implementation. However, when adopting uniform quantization techniques, it faces challenges in accurately capturing both the dense region and rare

Table 1: Property comparison of SuFP against MSFP and BSFP.

| Property | SuFP | MSFP | BSFP |
|---|---|---|---|
| Reflecting Diverse Density | ○ **(Piecewise)** | × (Uniform) | △ (Semi-Uniform) |
| Implementation Complexity | **Easy (Tensor-Wise)** | Difficult (Block-Wise) | Difficult (Block-Wise) |
| Adaptation to Both Activations and Weights | ○ | ○ | × (Only Weight Possible) |
| Hardware Efficiency | ○ | ○ | × (Bit-Serial) |

outliers. These challenges become more pronounced as the bit-width decreases. On the other hand, an approach of floating-point quantization becomes increasingly compelling, given that many neural network models exhibit Gaussian-like distributions in their weights and activations. This method offers superior flexibility in addressing diverse data distributions compared to INT quantization. However, floating-point quantization has its own challenges. One of the significant hurdles is the difficulty in preserving the precision of individual values due to its limited mantissa bit-width.

Another approach that has become increasingly adopted is piecewise quantization (Fang et al., 2020; Jain et al., 2019; Yuan et al., 2022). This method divides the quantization range into several distinct regions, tailoring the precision for each segment. The motivation behind this method stems from the need for an effective representation of both dense and sparse regions. However, piecewise quantization's drawback lies in the difficulty of achieving low errors across all regions. It's crucial to distribute the limited bit budget across regions carefully. Increasing the number of regions for more accurate representation leads to a trade-off: insufficient bits for each region, significantly impacting accuracy.

In the domain of block floating-point, multiple data points are grouped within a single block, sharing a unified exponent. Prominent examples of this approach are MSFP (Darvish Rouhani et al., 2020) and BSFP (Lo et al., 2022). In MSFP, the largest exponent of the block is used as "a shared exponent." Each mantissa is then right-shifted to align with this shared exponent. However, the mantissa is uniformly quantized, making it vulnerable to non-uniform data distributions in the block. In contrast, BSFP identifies the limitations of having a single shared exponent for each mantissa within a block and suggests an alternative approach. It introduces a method in which a single mantissa is split into two subwords, each sharing a scaling factor, adding finer granularity to the block's structure. However, BSFP is unsuitable for activation tensors due to the challenge of finding the optimal combination of two scalings in real-time, and its PE works in a bit-serial manner with 2-bit inputs or even 1-bit, leading to considerable hardware inefficiencies. In addition, block floating-point quantization inherently faces a trade-off between block size and model accuracy. Smaller blocks improve accuracy but increase the model size, while larger blocks decrease the model size but at a cost to accuracy.

In this paper, we introduce a groundbreaking data type, SuFP, effectively addressing the challenges discussed above. Table 1 summarizes the property comparison between SuFP and other major data types for quantization. In the following section, we will describe the details of SuFP.

## 3 SUPER FLOATING-POINT (SUFP)

The key idea of SuFP is multi-region piecewise quantization using a tensor-wise scalable bias. With this scheme, SuFP can accurately represent both dense and sparse regions in data distribution, ensuring a various dynamic range with high bit utilization.

**Multi-Region Piecewise Quantization.** SuFP data type comprises the sign bit (MSB), an encoding field, and a data field. SuFP can present three different data representations of exponent-and-mantissa combinations based on the encoding field, as shown in Figure 3, and the data field is interpreted according to each representation.

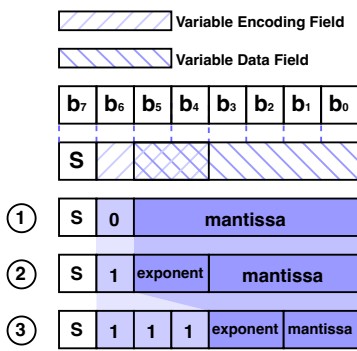

Figure 3: Visual representation of multi-region piecewise data format of SuFP.

**Algorithm 1:** SuFP Decoding Algorithm

**Input** : A Binary 8-bit Data $[b_7, b_6, \ldots b_0]$
**Output:** SuFP Quantized Data

1   $s_{sufp} \leftarrow b_7$
2   **if** $b_6 == 0$ **then**
3     $e_{sufp} \leftarrow -(100_2) \cdots \textcircled{1}$
4     $m_{sufp} \leftarrow b_5 b_4 b_3 b_2 b_1 b_0 \cdots \textcircled{1}$
5   **else**
6     **if** $b_5 b_4 \neq 11_2$ **then**
7       $e_{sufp} \leftarrow -(10_2) + b_5 b_4 \cdots \textcircled{2}$
8       $m_{sufp} \leftarrow 1 b_3 b_2 b_1 b_0 \cdots \textcircled{2}$
9     **else**
10       $e_{sufp} \leftarrow 11_2 + b_3 b_2 \cdots \textcircled{3}$
11       $m_{sufp} \leftarrow 1 b_1 b_0 \cdots \textcircled{3}$
12   **return** $(-1)^{s_{sufp}} \cdot m_{sufp} \cdot 2^{e_{sufp}}$

The overall decoding process is described in Algorithm 1. Once the representation is determined, exponent ($e_{sufp}$) and mantissa ($m_{sufp}$) are determined based on the data field. Meanwhile, to achieve different precisions, each representation has a distinct exponent baseline. The exponent baselines for representations ①, ②, and ③ are $-4$, $-2$, and $3$, respectively.

There are other features that influence the precision of each representation. The representation ① is designed to express numbers close to zero with high granularity by using the entire data field for the mantissa. Notably, by setting $b_5$ and $b_4$ to $00_2$, it achieves the representation of subnormal numbers in the IEEE floating-point standard. On the other hand, both the representation ② and ③ divide the data field into exponent and mantissa sections. Specifically, the representation ③ aims to capture a broader range of numbers, including outliers, even with its fewer mantissa bits compared to representation ②. In contrast, the representation ② focuses on numbers within an intermediate range between representation ① and ③. It offers finer precision due to its wider bit-width mantissa, allowing for an optimal expression of numbers between the main body and outliers.

Building on these specifics, each representation offers different levels of precision within its respective range, effectively covering a wide spectrum of numbers. For example, numbers in the range of 0 to 4, which belong to the main body, are captured with a granularity of $2^{-4}$. On the other hand, outliers in the range of 256 to 448 are captured with a granularity of $2^6$. Due to the implementation of the variable encoding field and variable data field, SuFP is capable of effectively representing both sparse and dense regions. For detailed experimental results and analysis, please refer to Appendix B and Appendix H.

**Tensor-wise Scalable Bias.** We introduce "tensor-wise" scalable bias. The scalable bias is the value added to the exponent baseline to determine the actual exponent value. By employing this approach, we can configure a more diverse range of exponent values than what can be originally represented with the limited exponent bit-width. More specifically, SuFP's scalable bias employs a 5-bit 2's complement, enabling an extension of the exponent range from -16 to +15 for each representation.

As shown in Appendix C, we capture that the tensor-wise distribution closely follows the channel-wise distribution, which lead us to employ scalable bias at the tensor level. As a result, SuFP enables precision tuning for various data distributions within the model, even while using larger tensor units instead of smaller block units. Furthermore, by adopting this tensor-wise approach, we have also gained the advantages of increased flexibility in computational direction within the PEs and reduction in memory footprint.

**Computation with SuFP.** Before beginning the computation, we need a bias-selecting process as detailed in Appendix A. Once the bias for each tensor is predetermined, a tensor is quantized as SuFP without the need to search for the optimal bias at runtime. This enables in-situ quantization for weight and activation tensors. Real-time quantization of the activation tensor is possible due to the highly consistent distribution of activation data during inference, regardless of input variations.

In the inference operations of SuFP, the SuFP Arithmetic Logic Unit (ALU) is utilized to execute multiplications on mantissa and additions on exponents, as described in Equation (1) and (2). The largest value from the exponent addition results is identified. To align all the results with the maximum exponent value, the results of the mantissa multiplications are right-shifted, as shown in Equation (3). This methodology is similar to the MSFP method used to determine a shared exponent in a block. The aligned mantissa are then processed through an adder tree to compute the partial sum. The summation of the largest exponent value and the scalable bias yields the partial result for the exponent, as detailed in Equation (4).

$$(\boldsymbol{w}_i \cdot 2^{\text{bias}_W}) \cdot (\boldsymbol{a}_i \cdot 2^{\text{bias}_A}) = \sum_{j=0}^{n-1} (-1)^{s_{w,j}} m_{w,j} 2^{e_{w,j}+bias_W} \cdot (-1)^{s_{a,j}} m_{a,j} 2^{e_{a,j}+bias_A} \quad (1)$$

$$= \sum_{j=0}^{n-1} (-1)^{s_{w,j} \oplus s_{a,j}} m_{w,j} m_{a,j} \cdot 2^{e_{w,j}+e_{a,j}} \cdot 2^{bias_W+bias_A} \quad (2)$$

$$(m_{w,j} m_{a,j})' = (m_{w,j} m_{a,j}) >> e_{max}, \; e_{max} = \max_{0 \le i \le n-1} (e_{w,i} + e_{a,i}) \quad (3)$$

$$(\boldsymbol{w}_i \cdot 2^{\text{bias}_W}) \cdot (\boldsymbol{a}_i \cdot 2^{\text{bias}_A}) \approx 2^{bias_W+bias_A+e_{max}} \cdot \sum_{j=0}^{n-1} (-1)^{s_{w,j} \oplus s_{a,j}} (m_{w,j} m_{a,j})' \quad (4)$$

**Hardware Implementation for SuFP.** Figure 4 shows the architecture of the proposed SuFP PE, designed to process 16 parallel input data streams. Since SuFP can effectively handle a wide dynamic range using scalable bias, SuFP PE is composed mainly of integer operation ALUs and shifters, requiring fewer hardware resources. Furthermore, scalable bias overhead in the PE is negligible as it is accommodated through a single integer addition with the maximum exponent value in the accumulator. The SuFP PE can be readily integrated into any architecture, with the systolic array being a prime example. For a detailed explanation of the systolic array, please refer to Appendix G.

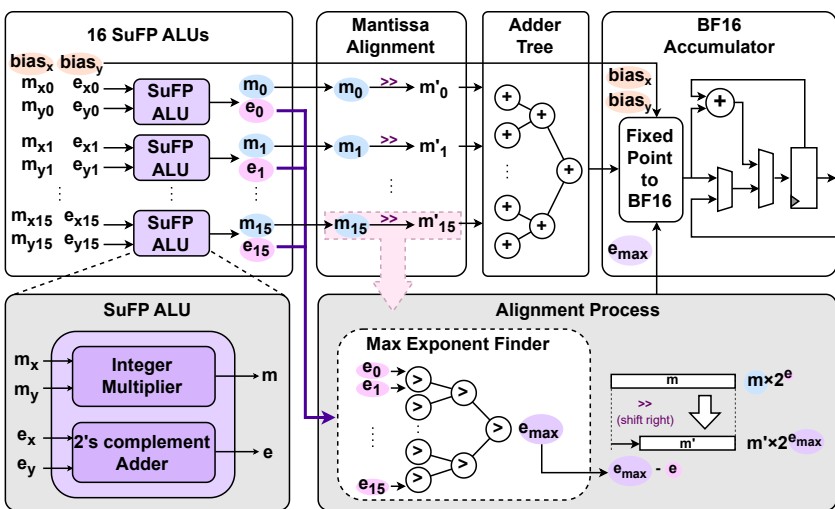

Figure 4: Proposed processing element architecture for SuFP.

## 4 EXPERIMENTS

This section evaluates the proposed SuFP described in Section 3. We comprehensively assess SuFP's performance by quantizing both weight and activation tensors across various models, including vision, language, and text-to-image. We then analyze the reduction in memory footprint overhead achieved by SuFP. By implementing SuFP PE, we demonstrate its performance and energy efficiency improvements.

## 4.1 BASELINES AND EXPERIMENTAL SETUP

We implement SuFP using PyTorch with HuggingFace transformer and TorchVision libraries. For image classification tasks, we benchmark our method on the ResNet18, ResNet50 (He et al., 2016), Vision Transformer (ViT) (Dosovitskiy et al., 2020), and EfficientNet-v2 (Tan & Le, 2021) models with the ImageNet dataset (Deng et al., 2009). For natural language tasks, we benchmark our method using the BERT-base model (Devlin et al., 2018) on datasets such as MRPC, CoLA (Warstadt et al., 2018), and SQuAD 2.0 (Rajpurkar et al., 2018). For text-to-image generative tasks, we benchmark our approach using the Stable Diffusion v2 (Rombach et al., 2021) on the COCO dataset (Lin et al., 2014). For LLMs, we benchmark our method using Llama 2 model Touvron et al. (2023) on MMLU. We compare the performance of the proposed SuFP with the baseline data types, including Floating-Point 32 (FP32), Floating-Point 16 (FP16), Brain Floating-Point 16 (BF16), Floating-Point 8 (FP8), MSFP, and BSFP.

Additionally, we use SystemVerilog to implement the SuFP PE and various baseline PEs, including FP32, BF16, FP8, MSFP, and BSFP. All designs are synthesized using the Synopsys Design Compiler, optimized for the 28nm CMOS technology, and set to operate at 500MHz clock frequency. In addition, we evaluate the power estimation of each PE with internal power, switching power and leakage power. The implementation details of the previously mentioned PEs are elaborated in Appendix F.

## 4.2 ACCURACY EVALUATION

We evaluate the impact on the accuracy of our SuFP on vision models. We compare its accuracy against standard formats like FP32, BF16 and FP8, as well as other PTQ techniques, including MSFP and BSFP. We use the configuration of MSFP and BSFP based on their reported superior accuracy in previous studies. Furthermore, we set the number of elements in a block to 16, as this value gives optimal performance for both MSFP and BSFP.

To highlight the adaptability of SuFP across various domains, we extend our experiments to models in the language and text-to-image areas. In this evaluation, we compare the performance of SuFP with that of FP16, FP8, MSFP and BSFP.

Table 2: Comparison of normalized accuracy among vision models with SuFP and other data types.

| Method | Data Type | Vision | | | |
|---|---|---|---|---|---|
| | Weight / Activation | ResNet-18 | ResNet-50 | EfficientNet-v2 (s) | ViT-B/16 |
| FP32 | FP32 / FP32 | 1.0000 (69.76/69.76) | 1.0000 (76.13/76.13) | 1.0000 (81.31/81.31) | 1.0000 (81.07/81.07) |
| BF16 | BF16 / BF16 | 1.0006 (69.80/69.76) | 0.9997 (76.11/76.13) | 1.0000 (81.31/81.31) | 0.9996 (81.04/81.07) |
| MSFP[1] | MSFP / MSFP | 0.9990 (69.69/69.76) | 0.9991 (76.06/76.13) | 0.9983 (84.09/84.23) | 0.9993 (81.01/81.07) |
| BSFP[2] | BSFP / MSFP | 0.9987 (69.67/69.76) | 0.9993 (76.08/76.13) | 0.9980 (84.06/84.23) | 0.9981 (80.92/81.07) |
| **SuFP** | **SuFP / SuFP** | **1.0007 (69.81/69.76)** | **1.0005 (76.17/76.13)** | **0.9994 (81.26/81.31)** | **0.9999 (81.06/81.07)** |

[1] The precision of MSFP is characterized as MSFP16 (1bit sign, 7bit mantissa, 8bit exponent).
[2] BSFP is structured with 5-bit and 2-bit mantissa, accompanied by 8-bit and 7-bit scale factors corresponding to each mantissa.

**Evaluation on Vision Models.** Table 2 compares the performance of various quantization techniques in the vision domain. The top-1 accuracy metric is used for performance evaluation. For a consistent comparison, we source the accuracy results for MSFP and BSFP from the BSFP paper. To ensure consistency, we set up our environment on FP32 accuracy from the BSFP paper. However, FP32 accuracy for EfficientNet-v2 differs from the reported value. Thus, we normalize the accuracies of MSFP, BSFP, and SuFP relative to FP32 and focus on comparing their changes.

As demonstrated in Table 2, when SuFP is applied, the overall accuracy drop is negligible. For instance, in EfficientNet-v2, MSFP and BSFP have an accuracy drop of 0.17% and 0.2% compared to full-precision, respectively. In contrast, SuFP shows only a 0.06% decrease in accuracy.

**Evaluation on Language and Text-to-Image Models.** Table 3 shows the performance of SuFP on the Language and Text-to-image categories. For comparison, we use FP16, FP8, MSFP, and

Table 3: Comparison of performance among language and text-to-image generative models with SuFP and other data types.

| Method | Data Type | BERT-base | | | Stable Diffusion v2 |
|---|---|---|---|---|---|
| | Weight / Activation | MRPC ↑ (Accuracy) | CoLA ↑ (MCC) | SQuAD 2.0 ↑ (F1-score) | COCO ↓ (FID-score) |
| FP16 | FP16 / FP16 | 0.8307 | 0.5678 | 78.8684 | 27.0643 |
| FP8[1] | FP8 / FP8 | 0.6516 | 0.5241 | 56.6289 | 459.9799 |
| MSFP | MSFP / MSFP | 0.8319 | 0.5636 | 78.8113 | 27.2551 |
| BSFP | BSFP / MSFP | 0.8336 | 0.5636 | 78.7647 | - |
| **SuFP** | **SuFP / SuFP** | **0.8371** | **0.5756** | **78.9547** | **25.6262** |

[1] E4M3 format is used for FP8 (1bit sign, 4bit mantissa, 3bit exponent).

**BSFP as baseline.** In the Language category, BERT-base serves as our representative model. We evaluate the BERT-base model on the MRPC, CoLA, and SQuAD 2.0 datasets using accuracy, MCC, and F1-Score as the performance metrics, respectively. Based on the experimental results, SuFP demonstrates better results across various benchmarks for the BERT-base model.

For the Text-to-image category, we experiment with the Stable Diffusion v2 model. We use the COCO dataset in this experiment, adopting the FID score as our performance metric. For the experiment, we adopted our SuFP only to diffusion. In our experimental results, SuFP achieves an FID score of 25.6262, showing improved performance over FP16. Also, we obtain images of equal or better quality, please refer to Appendix I for the examples of generated images.

Table 4: Comparison of performance among LLMs with SuFP and other data types.

| Method | Data Type | Llama 2-7b | | | | |
|---|---|---|---|---|---|---|
| | Weight / Activation | STEM | Humanities | Social Sciences | Other | Average |
| FP16 | FP16 / FP16 | 0.369 | 0.433 | 0.518 | 0.525 | 0.459 |
| FP8 | FP8 / FP8 | 0.214 | 0.242 | 0.217 | 0.238 | 0.229 |
| MSFP | MSFP / MSFP | 0.372 | 0.431 | 0.523 | 0.524 | 0.460 |
| **SuFP** | **SuFP / SuFP** | 0.349 | 0.403 | 0.486 | 0.493 | 0.430 |

**Evaluation on LLMs.** Table 4 shows the performance of SuFP on the LLMs. For comparison, we use FP16, FP8 and MSFP as baseline. We evaluate the Llama 2 model on MMLU benchmark. Based on the experimental results, SuFP shows marginal reductions in performance for the Llama 2 model. Despite using the same bit precision, SuFP achieves better performance compared to FP8.

## 4.3 MEMORY FOOTPRINT

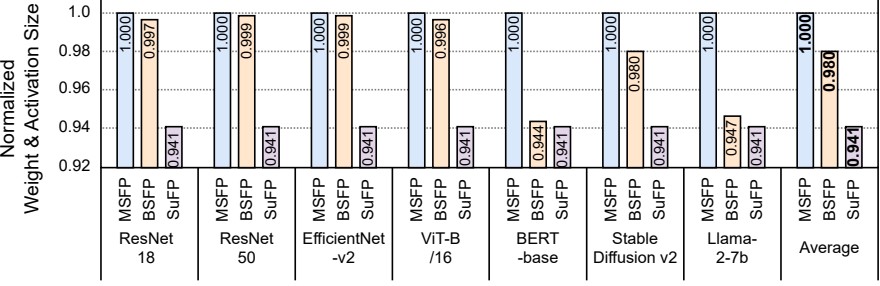

Figure 5: Memory footprint of MSFP, BSFP and SuFP normalized to MSFP.

Figure 5 shows the required memory footprint for the weight and activation tensors of models applied with the quantization techniques of SuFP, MSFP, and BSFP. They are normalized with respect to MSFP to illustrate the effective reduction in footprint clearly. It is worth noting that BSFP uses 128-bit memory due to the standard byte alignment despite its 127-bit configuration, causing 0.8% overhead. The detailed methodology for calculating the memory footprint for each data type is provided in Appendix E. Based on the calculations, SuFP occupies 0.941× of the memory of MSFP and 0.960× that of BSFP on average. These results confirm that SuFP outperforms other methods in compression capability.

## 4.4 Hardware Efficiency

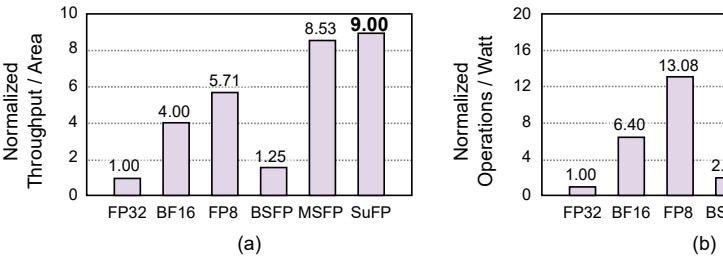

Figure 6: (a) Normalized throughput per area and (b) normalized operations per watt comparison of SuFP with other baselines. The values are normalized with respect to FP32.

Finally, we compare the hardware efficiency of SuFP with that of other methods. Figure 6 (a) shows the throughput per area of PEs for various data formats, normalized with the result of FP32. The SuFP PE demonstrates the highest efficiency, 9.00× compared to FP32 PE. On the other hand, BSFP shows a lower value due to its use of bit-serial PE. The bit-serial operation in BSFP PE necessitates 16 computations to process the multiplication of a 7-bit BSFP mantissa with a 7-bit MSFP mantissa using a 2-bit multiplier. In the case of MSFP, while MSFP utilizes a 7-bit multiplier to multiply mantissa, SuFP employs a 6-bit multiplier, enabling SuFP PE to achieve enhanced throughput-per-area efficiency. In more detail, SuFP PE is up to 7.20× more efficient than that of state-of-the-art MSFP and BSFP PE. This clearly demonstrates that SuFP has the most compact hardware structure compared to other data types.

We also evaluate the performance of SuFP PE in terms of energy efficiency. Figure 6 (b) shows the comparison results, presenting the number of operations per watt, which is normalized to FP32. SuFP outperforms FP32 PE by being 17.04× more energy-efficient. Even when compared to the state-of-the-art MSFP and BSFP, our PE is up to 8.27× more energy-efficient. Furthermore, the SuFP PE achieves significantly higher accuracy and far superior hardware efficiency despite using the same number of bits as FP8 PE. Through these results, we can once again demonstrate that SuFP is not only an accurate quantization method but also hardware-efficient. Additionally, the area and the power values of various PEs for the two experiments in this section are provided in Appendix F.

## 5 Conclusions

This paper introduces Super Floating-Point, designed to tackle the challenges of large and complex DNNs. SuFP combines multi-region piecewise quantization and tensor-wise scalable bias, optimizing precision for different data regions. This method captures dense near-zero data and outliers, adapting to diverse data distributions. The tailored SuFP PE uses only integer units and shifters for compactness. Experimentally, SuFP not only matches or surpasses FP32 in accuracy but also enhances computational capability and energy efficiency by 9.00× and 17.04× over FP32. Furthermore, its computational capability and energy efficiency outperform state-of-the-art MSFP and BSFP by up to 7.20× and up to 8.27×, respectively.

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

## APPENDIX A  QUANTIZATION FLOW FOR SUFP

---

**Algorithm 2:** Scalable bias-optimal quantization flow

---

**Input**  : Model $M$, Bias searching range $[b_i, b_j]$,
 Full-precision weight $W_{fp}$, Full-precision Activation $A_{fp}$
**Output:** Optimal scaling weight bias set $b_{w\_opt}[0 \dots n-1]$,
 Optimal scaling activation bias set $b_{a\_opt}[0 \dots n-1]$

1  // Optimize weight bias set
2  **for** $layer \leftarrow 0\ to\ n-1$ **do**
3     $accuracy_{max} \leftarrow 0$
4     $W_{quant}[0 \dots layer-1] \leftarrow Quant(W_{fp}[0 \dots layer-1], b_{w\_opt}[0 \dots layer-1])$
5     **for** $bias = b_i\ to\ b_j$ **do**
6        $W_{quant}[layer] \leftarrow Quant(W_{fp}[layer], bias)$
7        $W' \leftarrow \{W_{quant}[0 \dots layer], W_{fp}[layer+1 \dots n-1]\}$
8        $accuracy \leftarrow Test(M, W', A_{fp})$
9        **if** $accuracy > accuracy_{max}$ **then**
10          $accuracy_{max} \leftarrow accuracy$
11          $b_{w\_opt}[layer] \leftarrow bias$

12 // Optimize activation bias set
13 **for** $layer \leftarrow 0\ to\ n-1$ **do**
14    $accuracy_{max} \leftarrow 0$
15    $A_{quant}[0 \dots layer-1] \leftarrow Quant(A_{fp}[0 \dots layer-1], b_{a\_opt}[0 \dots layer-1])$
16    **for** $bias \leftarrow b_i\ to\ b_j$ **do**
17       $A_{quant}[layer] \leftarrow Quant(A_{fp}[layer], bias)$
18       $A' \leftarrow \{A_{quant}[0 \dots layer], A_{fp}[layer+1 \dots n-1]\}$
19       $accuracy \leftarrow Test(M, W_{quant}, A')$
20       **if** $accuracy > accuracy_{max}$ **then**
21          $accuracy_{max} \leftarrow accuracy$
22          $b_{a\_opt}[layer] \leftarrow bias$

23 **return** $b_{w\_opt}, b_{a\_opt}$

---

The scalable bias of SuFP is set as a single value within each layer. In addition, once the bias is determined, it remains invariant throughout the entire inference process of the model. The bias determines the quantization range and precision, significantly impacting the overall accuracy. Thus, employing a proper bias for optimization is extremely important.

In seeking the tensor-wise optimal bias, there are potential risks of slipping into local optimization instead of achieving global optimization. Concurrently, individual optimal bias for weight and activation might not yield the best outcome when computed together.

Based on these insights, we structure our optimization process as described in Algorithm 2. This process considers (i) interactions among adjacent layers, (ii) cumulative influences across the layers, and (iii) the synergistic relationship between weight and activation. Additionally, the total time required for this procedure depends on the target accuracy level, which can dynamically change the bias searching range.

## APPENDIX B  EFFECTIVENESS OF MULTI-REGION PIECEWISE QUANTIZATION

In this section, we discuss the effect of multi-region piecewise quantization applied to SuFP. As shown in Figure 7, Gaussian distribution can be divided into a dense region in the near-zero range and a sparse region with rare large values. The dense region contains most of the data and the sparse region consists of values that significantly impact model accuracy. Therefore, accurately representing both regions is essential to minimize model performance degradation due to quantization.

In this experiment, we analyze the performance of piecewise quantization in both dense and sparse regions. The effect depends on the location and number of boundaries that divide regions, which means when setting the boundaries, it is crucial to accurately ensure the granularity required by

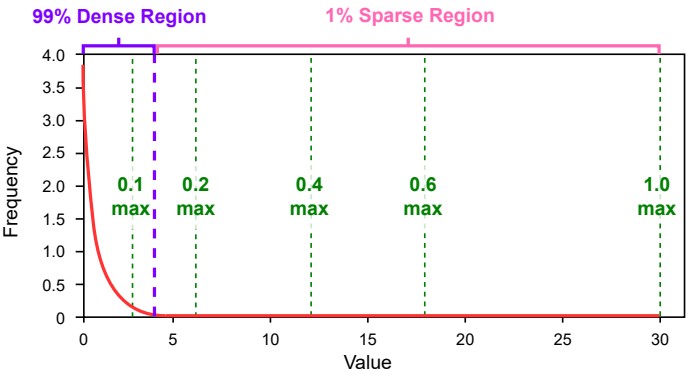

Figure 7: Real data distribution and piecewise quantization boundaries.

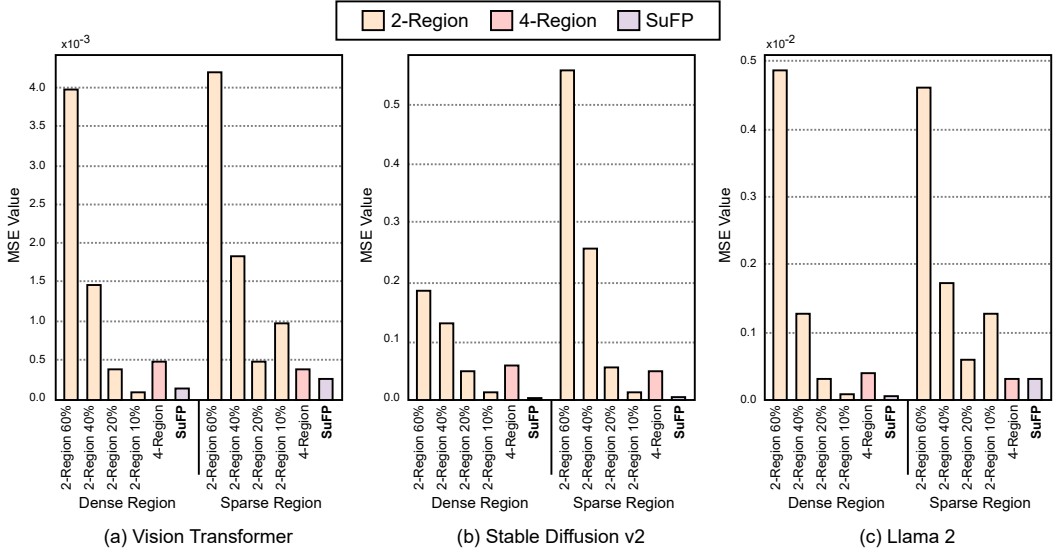

Figure 8: MSE value comparison of SuFP and other piecewise quantization techniques.

each region to minimize accuracy degradation in the model. Therefore, in this experiment, we set four boundaries at 10%, 20%, 40%, and 60% for 2-region quantization and measure the mean square error (MSE) in both dense and sparse regions for each setting. In the 4-region piecewise quantization experiment, we segmented the data into four distinct regions. This segmentations are determined by the three boundaries that demonstrated the lowest MSE in the previous 2-region quantization experiment. The models used in this experiment are Vision Transformer, Stable Diffusion v2 and Llama 2, and the dense and sparse regions are divided into 99% and 1% of the total data region, respectively.

To perform in-situ quantization and evaluate the accuracy of quantization process, we set a specific batch size for each model and obtain the maximum value within a batch to determine a scale factor. By using the scale factor, quantization is performed independently on each tensor within every batch. Subsequently, we calculate the MSE between the original and quantized tensors within the dense regions and the sparse regions.

As shown in Figure 8, achieving low MSE values in both the sparse region and the dense region is challenging in the case of 2-region piecewise quantization. On the other hand, 4-region quantization exhibits relatively consistent MSE values but higher in both of the regions. In the case of n-bit piecewise quantization with $N$-regions, the data bit-width allocated to each region is fixed to $n -$

$log_2(N) - 1$. This means that even if the number of regions is increased to represent the entire data distribution more finely, the granularity for each region is suboptimal. This issue becomes a hurdle in accurately representing the entire region including both sparse and dense regions. Consequently, this inflexible way of allocating bits leads to lower the overall accuracy of the model.

In contrast, SuFP consistently shows the lowest MSE values across all models and data regions. This is because SuFP ensures optimized granularity for each data region. Specifically, as explained in Section 3, SuFP uses a variable encoding field and variable data field to represent the data distribution, allowing for sufficient granularity for the high-granularity-required dense region while maintaining extensive dynamic range for the sparse region demanding wide coverage. This approach effectively represents the Gaussian distribution across each region, minimizing degradation in model performance.

## APPENDIX C COMPARATIVE ANALYSIS OF TENSOR-WISE AND CHANNEL-WISE QUANTIZATION

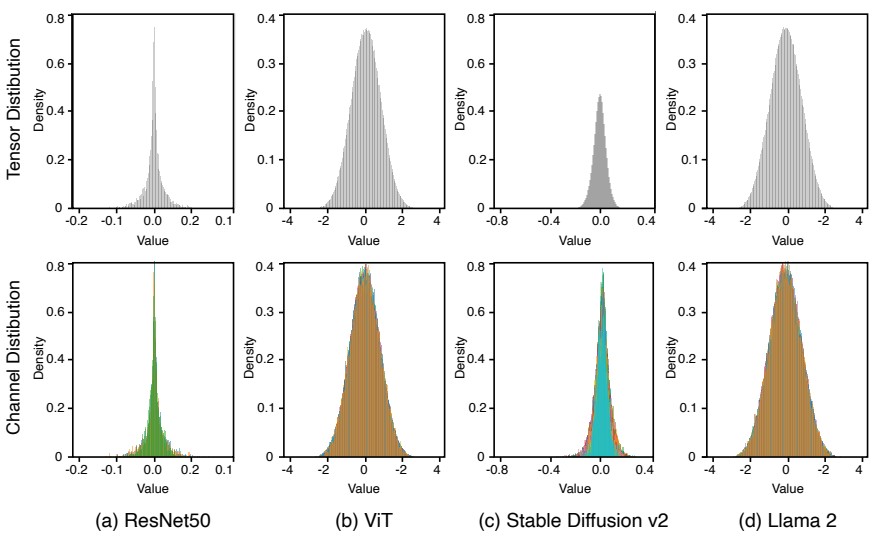

Figure 9: Distribution of each channel represented in different colors (bottom) and distribution of the overall tensor (top)

In this section, we analyze the channel-wise and tensor-wise quantization approaches. Quantization is commonly applied using a channel-wise approach, as data distribution clustering often occurs at the channel level. Such technique enables quantization that reflects the data distribution of each channel, effectively capturing data characteristics. On the other hand, it is anticipated that employing a tensor-wise approach may not correspond as closely with data characteristics as the channel-wise method. Nevertheless, this approach offers the advantages of flexible computational direction within the processing element and reduced memory footprint.

Figure 9 illustrates the data distribution in various models when employing tensor-wise and channel-wise approaches, showing the distribution for each channel with different colors in the latter case. Through this figure, we can observe similarities in the data distribution of the entire tensor and that of each constituent channel. These similarities include comparable patterns of outliers, overall dynamic ranges, and distribution shapes.

Based on these analyses, we choose the tensor-wise approach. This decision enable us to capture data characteristics at a similar level to the channel-wise approach, while also achieving advantages in flexible computational direction and memory usage.

## APPENDIX D    ROBUSTNESS OF SuFP TO OUTLIER

In this section, we analyze the impact of outlier increases on quantization within the activation layers of neural networks. Outliers, often occurring in activation layers, can significantly affect the quantization process, making them critical for the accuracy of the model. Therefore, it is necessary to evaluate the robustness of quantization methods to outliers.

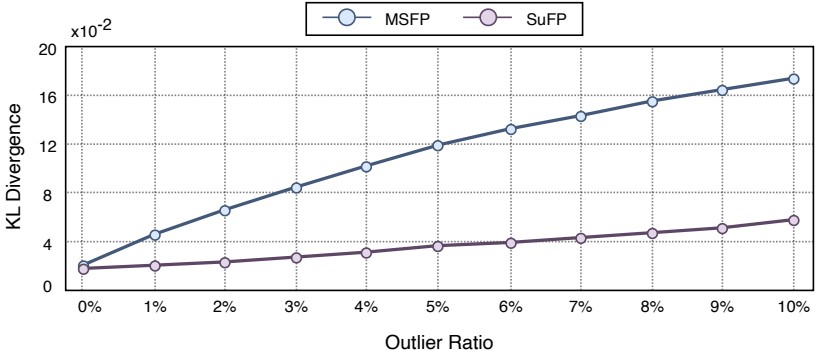

Figure 10: Comparison of Kullback-Leibler divergence for SuFP and MSFP quantization methods under varying outlier ratio.

In this study, we progressively increased the proportion of outliers in the tensor from 0% to 10% following a Gaussian distribution and measured the Kullback-Leibler (KL) divergence between original tensors and quantized tensors. Figure 10 shows the experimental results for two quantization methods, SuFP and MSFP. This experiment was repeated 1,000 times to obtain average values. BSFP was also considered; however, since it utilizes MSFP for activation quantization, it was only compared with MSFP.

Figure 10 demonstrates that as the proportion of outliers increases, the tensor quantized with SuFP shows a relatively lower KL divergence than the tensor quantized with MSFP. MSFP, which represents values using only the mantissa within a block, may incur some information loss when outliers are present. In contrast, SuFP, employing tensor-wise scalable bias and multi-region piecewise quantization, effectively represents both outliers and other values. These results indicate that SuFP is more robust against outliers compared to MSFP.

## APPENDIX E    CALCULATION METHODS FOR MEMORY FOOTPRINT

The calculation methods for the memory footprint per tensor of SuFP, MSFP, and BSFP are as follows.

**Memory Footprint of SuFP.**    Both activation and weight tensors are quantized with SuFP. SuFP quantizes each element into an 8-bit value. Additionally, a single 5-bit bias value per tensor is used. The overall memory footprint for SuFP is computed as:

$$E \times 8 + 5 \tag{5}$$

In this equation, $E$ represents the number of elements per tensor.

**Memory Footprint of MSFP.**    We use MSFP16 configuration, which has shown the best accuracy. This format quantizes each element into an 8-bit value. Additionally, it contains a single 8-bit shared exponent value per block. The block size of 16 elements is determined based on the best performance. Quantization is performed for both activations and weights using MSFP. The overall memory footprint for MSFP is computed as follows:

$$B \times (16 \times 8 + 8) = B \times 136 \tag{6}$$

$B$ denotes the number of blocks per tensor. Hence, MSFP occupies 136-bit per block.

**Memory Footprint of BSFP.** Since activation values are not fetched offline, BSFP cannot quantize them on the fly. Thus, BSFP quantizes only weights, while MSFP handles activations. As previously described, we select MSFP16 configuration for MSFP. For BSFP, we adapt an optimal 7-bit subword structure (5-bit + 2-bit). Each element comprises two subwords: 5-bit and 2-bit, totaling 7-bit. Moreover, BSFP includes 8-bit and 7-bit two scaling factors per block. Like MSFP, a block consists of 16 elements. The number of bits of BSFP used per tensor is:

$$B \times (16 \times (5 + 2) + (8 + 7)) = B \times 127 \tag{7}$$

Therefore, BSFP requires 127 bits of memory per block. However, in standard memory architectures, data storage sizes are typically in powers of two. Consequently, when a block quantized by BSFP is stored in real memory, 1-bit zero padding is added, allowing it to be stored as 128-bit. The overall memory footprint for BSFP is computed as follows:

$$B \times (16 \times (5 + 2) + (8 + 7) + 1) = B \times 128 \tag{8}$$

Thus, BSFP essentially occupies 128 bits per block. The overhead incurred from zero padding is just 0.8%, which is reasonable given the constraints of standard memory architectures. Figure 5 presents the overall results calculated from each data type's memory footprint. From these results, it is evident that SuFP has superior memory efficiency.

## APPENDIX F    IMPLEMENTATION DETAILS OF VARIOUS PES

Table 5: Iso-throughput area and power of SuFP and other baselines.

|  | 16x FP32 | 16x BF16 | 16x FP8 | 16x BSFP | MSFP | SuFP |
|---|---|---|---|---|---|---|
| Area ($\mu m^2$) | 29731.9679 | 7430.9759 | 5205.3120 | 23829.1200 | 3485.7900 | 3303.7200 |
| Power ($mW$) | 19.8528 | 3.1024 | 1.5173 | 9.6144 | 1.2051 | 1.1649 |

In Section 4.4, we conducted experiments to compare the compactness and energy efficiency of SuFP PE with other PEs. For these experiments, we implement SuFP PE and various baseline PEs (FP32, BF16, FP8, MSFP, and BSFP) using SystemVerilog and synthesize them using the Synopsys Design Compiler in 28nm CMOS technology. By using this setup, we measure the area and power values of the PEs. For a clearer comparison, the results are presented in Table 5 with the same throughput value for each PEs. To provide further understanding of these results, the detailed configurations for each PE are described as follows:

- **FP32 PE** supports full precision FP32 operations, involving multiplication and accumulation operations in FP32 format for precise outcomes.

- **BF16 PE and FP8 PE** perform multiplication operations in BF16 and FP8 formats, respectively. Additionally, BF16 format accumulator is used in both BF16 PE and FP8 PE to ensure consistent accuracy.

- **MSFP PE**, configured with a block size of 16, performs 16 pairs of MSFP multiplication operations in parallel. MSFP PE also utilizes BF16 accumulator for partial sum calculations. Additionally, we use the MSFP configuration that achieves the optimal accuracy.

- **BSFP PE** performs 16 sets of 2-bit multiplications, operating in a bit-serial manner. BSFP PE also utilizes BF16 accumulator for partial sum calculations. Additionally, we use the BSFP configuration that achieves the optimal accuracy.

- **SuFP PE** conducts 16 SuFP multiplication operations in parallel, as shown in Figure 4. SuFP PE also utilizes BF16 accumulator, similar to the other PEs.

## APPENDIX G    EXTENSION OF SUFP PE TO SYSTOLIC ARRAY

A systolic array architecture consists of multiple PEs arranged in a 2D array format, allowing parallel data processing. This architecture is not only adopted by Google's TPU (Jouppi et al., 2021)

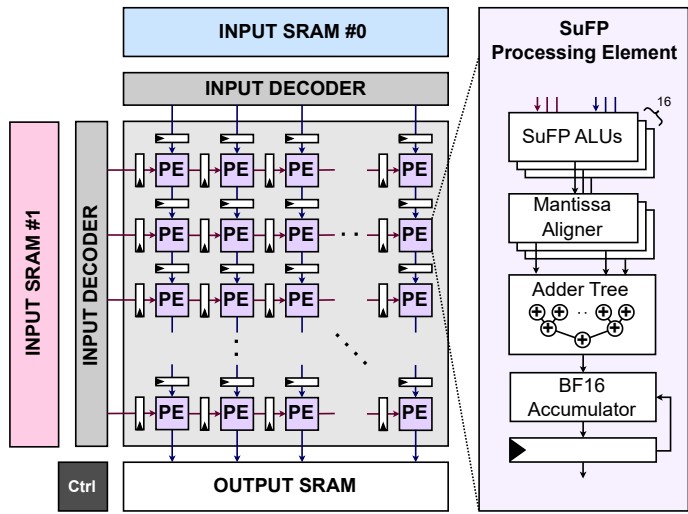

Figure 11: Systolic array architecture containing proposed SuFP PEs.

but is also widely utilized across various NPUs such as (Venkataramani et al., 2021; Geva et al., 2022; Gomes et al., 2022). Our proposed SuFP PE can also seamlessly integrate into the systolic array architecture, potentially leading to significant performance enhancements. Figure 11 illustrates the SuFP PE (on the right) and the systolic array architecture composed of these SuFP PEs (on the left). The PEs adjacent to the SRAM directly receive the decoded data from the SRAM. Subsequently, these PEs use this data for computations and transmit the computed results and input data to neighboring PEs. Through this process, once-decoded data efficiently propagates among PEs, thus minimizing the decoding-related overhead in the systolic array.

## APPENDIX H    ANALYSIS OF SuFP REPRESENTATIONS' IMPACT ON SuFP PE BIT UTILIZATION

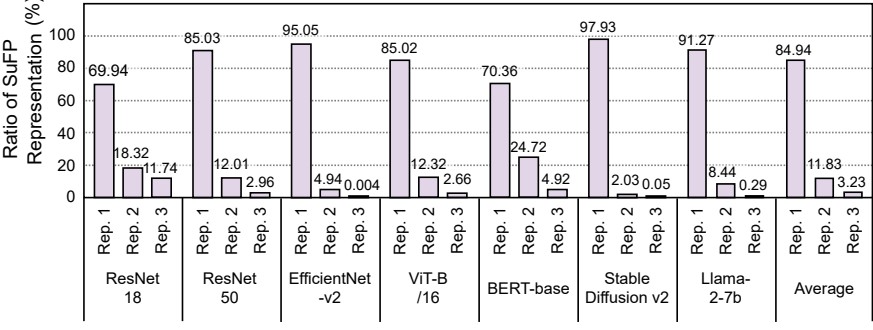

Figure 12: Comparison of SuFP representation proportion in weight and activation tensors across various models.

In this session, we analyze the impact of three different representations of SuFP on bit utilization in SuFP PE. Each representation has varied mantissa and exponent bit-widths, enabling optimized bit allocation in both dense and sparse regions. Representation ① has the largest mantissa bit-width, while representation ③ has the smallest. These differences also affect the design of SuFP PE ALU. The bit-width of SuFP PE ALU is set based on representation ① , which has the largest mantissa bit-width. Although representation ③ is primarily utilized to effectively represent outliers, it results in relatively lower bit utilization within SuFP PE.

Figure 12 shows the ratio of the three representations when applying SuFP quantization across various models. As shown in Figure 12, the ratio of representation ③ is noticeably low in all models. For instance, in EfficientNet-v2, representation ③ accounts for only about 0.004%. In conclusion, while representation ③ is essential for model accuracy, its low representation ratio indicates a minor impact on SuFP PE hardware's bit utilization. Consequently, SuFP demonstrates its effectiveness in terms of both accuracy and hardware's bit utilization.

## APPENDIX I    QUANTIZATION RESULTS ON TEXT-TO-IMAGE GENERATION

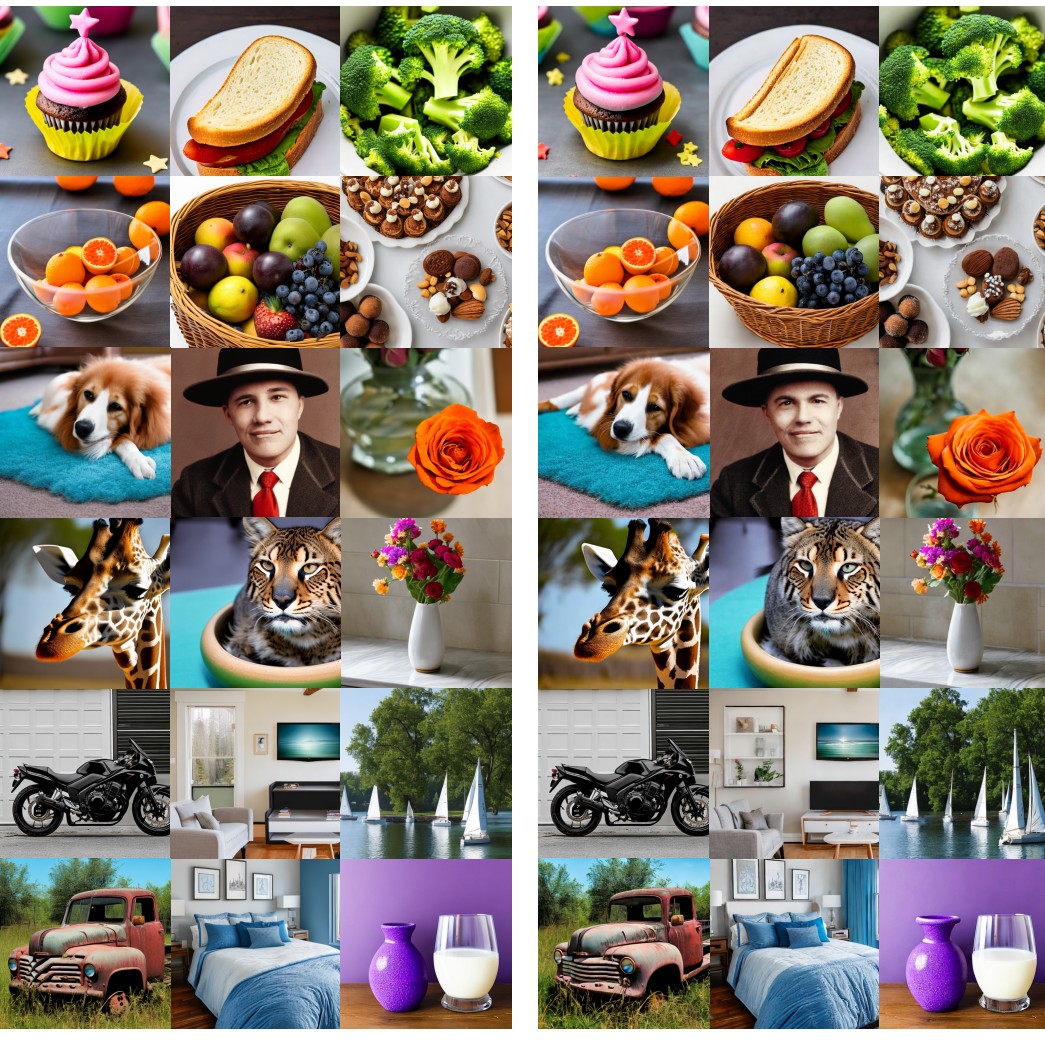

(a) Full Precision                                      (b) SuFP

Figure 13: Sample images generated from Stable Diffusion model on COCO dataset with full precision and our SuFP.

In this section, we provide the results of text-to-image generation using SuFP quantization applied to full-precision diffusion models. As shown in the figures below, there is almost no difference between images generated with full-precision and those produced using SuFP quantization.

