# OpenReview forum: "Super Floating-Point (SuFP): Efficient To All. Multi-Region Piecewise Quantization using Scalable Bias with Hardware Optimization"
_ICLR.cc/2024/Conference — Submitted to ICLR 2024_

### Official Review · Reviewer_gb1N · 2023-11-01

**Soundness:** 3 good
**Presentation:** 3 good
**Contribution:** 2 fair
**Rating:** 5
**Confidence:** 4

**Summary:**

This work introduces the super floating-point format (SuFP) which uses a piece-wise quantizer that better fits the standard distributions shown in modern deep neural networks. It defines these three modes for representing three regions of values and demonstrates that they outperform recently proposed floating-point formats. They evaluate across vision, language, and text-to-image models and show lower memory, area, and power than other approaches.

**Strengths:**

Figure 3 and Figure 4 clearly show the different data modes and hardware outlines.

This work includes evaluations across vision, language, and text-to-image models.

This work also evaluates the efficiency of its method across many performance categories.

**Weaknesses:**

Form 3 seems strange since by its bitwidth alone it seems to be strictly worse than Form 3. Does the different bias matter here?

The advantage of supporting difficult modes is not clear when the PEs seem to need to process the largest exponent and mantissas. It might be useful to show where the coverage is for each mode in a figure similar to Figure 2.

What is the definition for exponent baseline? Initially I thought the exponent would be added to these but the baseline for mode 2 includes the exponent bits themselves.

The paper seems to over-sell itself in many places and that space could be used to add more detail.

-- Minor --
It would be clearer to show the bitwidth in the formats in the tables since these comparisons are across 32, 16, and 8 bits.

Equation 1 can could be explained better in the text. For example, are the X elements an arbitrary tensor? It is unclear what the purpose of this equation is overall

Equations 2-5 could be aligned to make them more readable.

**Questions:**

Are there any additional floating-point quantization scale factors with this method? FP8 method often still use additional higher-precision quantization scales still.

What is the typical distributions of the SuFP modes? To justify sacrificing the bitwidth to handle different modes, it would be useful to see how often the modes are needed. The encoding seems to be a variable-bitwidth encoding so does it reflect the mode distribution? Does mode 3 show up the least often?

What FP8 variant is used for comparison? E4M3 typically has the highest accuracy.

Why not show all the formats for each category evaluation? Formats like BF16 should be simple to evaluate in the PyTorch setup for each and there is significant room in the tables.

Why is the BSFP datatype only 7 bits in Figure 2 while the others are 8 bits?
What granularity does the method operate at? For example, the bias is shared per tensor but since distributions cluster in channels why not make it shared per channel? The hardware diagram seems like there can potentially be bias per 16 elements. Is this true? Also, it seems possible to share the mode over a block of data depending on the variation there.

The ALU seems like it is purely doing standard floating-point multiplication? Or does it support additional functionality to justify its name?

---

> ### Author Response · Authors · 2023-11-23
> **Thanks for your valuable feedback!**
>
> We would like to extend my sincerest gratitude for your detailed and thorough review. Your meticulous analysis and insightful comments have significantly contributed to enhancing the quality and depth of our research. We have carefully reviewed the points you raised and, accordingly, conducted additional experiments and analyses to enhance the depth and quality of the paper. We would like to acknowledge that your attention to detail and the constructive feedback you offered have significantly contributed to improving the quality of our paper.
>
> **Your Questions**
> > - Form 3 seems strange since by its bitwidth alone it seems to be strictly worse than Form 3. Does the different bias matter here?
> > - The advantage of supporting difficult modes is not clear when the PEs seem to need to process the largest exponent and mantissas. It might be useful to show where the coverage is for each mode in a figure similar to Figure 2.
> > - What is the typical distributions of the SuFP modes? To justify sacrificing the bitwidth to handle different modes, it would be useful to see how often the modes are needed. The encoding seems to be a variable-bitwidth encoding so does it reflect the mode distribution? Does mode 3 show up the least often?
>
> **Q1.** **Consideration and discussion of various representations supported by SuFP.**
>
> **A1.** The multifaceted perspective you provided on the representations of SuFP has significantly contributed to clarifying the logical development of our research. Here is a detailed response.
>
> SuFP incorporates three different representations, each designed with distinct purposes. Representation 1 uses a small exponent baseline and a large bit-width for the mantissa, primarily to accurately depict the dense region around zero. In contrast, Representation 3 utilizes a large exponent baseline and exponent bit-width to capture sparse and wide-ranging outliers effectively. To empirically validate the effectiveness of our SuFP, we conducted experiments representing the dense and sparse regions of various data using SuFP's multi-region piecewise quantization approach. The details of these experiments are described in Appendix B. We evaluated the accuracy of our approach by comparing the mean square error before and after quantization, confirming that SuFP effectively represents data in each region.
>
> As you pointed out, we agree that there is a reduction in bit utilization due to the representation 3 in SuFP. This is because the processing element of SuFP is designed based on the representation with the largest bit-width. However, as confirmed through experiments, Representation 3, targeting outliers, is very rare in actual models. We applied SuFP to various models and analyzed the proportion of each representation, detailed in Appendix H. Therefore, the impact of Representation 3 on bit-utilization is negligible, and SuFP can effectively represent data while efficiently maintaining the utilization of the processing element.
>
> |                      | **ResNet18** | **ResNet50** | **EfficientNet-v2** | **ViT-B/16** | **BERT-base** | **Stable Diffusion v2** |**Llama-2-7b**| **Average** |
> |:--------------------|:------------:|:------------:|:-------------------:|:-------------:|:---------------:|:-------------------------:|:-------------:|:---------:|
> | **Representation 1 (%)** |     69.94    |     85.03    |        95.05        |     85.02    |     70.36     |          97.93          |    91.27    |  84.94 |
> | **Representation 2 (%)** |     18.32    |     12.01    |         4.94        |     12.32    |     24.72     |           2.03          |    8.44    |  11.83 |
> | **Representation 3 (%)** |     11.74    |     2.96     |        0.004        |     2.66     |      4.92     |           0.05          |     0.29    | 3.23 |

---

> ### Author Response · Authors · 2023-11-23
> **Thanks for your valuable feedback! (cont.)**
>
> **Your Question**
> > What granularity does the method operate at? For example, the bias is shared per tensor but since distributions cluster in channels why not make it shared per channel?
>
> **Q2.** **Considerations on Representation Sharing within SuFP PE**
>
> **A2.** As you have pointed out, and as illustrated in Figure 4, our SuFP PE architecture indeed allows for the application of specific biases every 16 elements. This design enables the sharing of identical representations for each PE input unit.
>
> However, this approach requires a decision to be made in cases of outliers: whether to increase granularity within that unit to represent the outlier accurately or to decrease it in favor of representing the majority of values. Related to this, our experiments detailed in Appendix D, involving the gradual addition of outliers to a Gaussian-distributed dataset, showed that SuFP can represent numbers through various representations within a single unit, maintaining robustness even with an increase in outliers. In contrast, MSFP, which relies solely on the mantissa for data representation, exhibited a significant increase in KL divergence. Based on these experimental results, we concluded that sharing representations across PE input units can potentially compromise robustness against outliers.
>
> -------
>
> **Your Question**
> > The hardware diagram seems like there can potentially be bias per 16 elements. Is this true? Also, it seems possible to share the mode over a block of data depending on the variation there.
>
> **Q3.** **Exploration of Scalable Bias Sharing Scope**
>
> **A3.** Your suggestion of sharing bias at the channel level is indeed interesting. We have also deeply considered this aspect. To explore the most effective scope for bias sharing, as detailed in Appendix C, we conducted experiments to compare the impacts of bias sharing at both tensor and channel levels. Our experimental results revealed that bias sharing at the tensor level exhibited data distributions similar to that at the channel level. Furthermore, sharing bias at the tensor level provided more flexibility in computation within SuFP and offered the additional benefit of reducing model size. Considering these experimental results, we decided to implement bias sharing at the tensor level.
>
> ------
> **Your Question**
> > Why is the BSFP datatype only 7 bits in Figure 2 while the others are 8 bits?
>
> **Q4. Inquiry Regarding BSFP Configuration**
>
> **A4.** You asked about the data type configuration of BSFP. In our study, we employed the configuration that demonstrated the best performance as reported in the BSFP paper, which includes both subword bit-width and scaling factor bit-width configurations. Our choice of this specific configuration is driven by the significant impact that these parameters have on the performance of quantization. Consequently, we adopted the configuration that exhibited superior performance among various configurations mentioned in the BSFP paper. This approach ensures the optimization of our research results and allows for a more accurate and fair comparison with BSFP.
>
> -------
> **Your Question**
> > Why not show all the formats for each category evaluation? Formats like BF16 should be simple to evaluate in the PyTorch setup for each and there is significant room in the tables.
>
> **Q5. Additional Experiments on the other format**
>
> **A5.** In response to your suggestions, we have extended our experiments beyond BF16 by directly implementing MSFP and BSFP using PyTorch. We conducted additional experiments to compare their performance directly with SuFP. The results of these performance comparisons can be found in Section 4. As shown in the table below, SuFP demonstrates generally impressive performance across both language and text-to-image models compared to the others. This table can be found in Table 3 in Section 4.
>
> | **Model**               | **Benchmark** | **Metric** | **MSFP** | **BSFP** | **SuFP** |
> |-------------------------|---------|-----------|:--------:|:--------:|:--------:|
> | **BERT-base**           |    MRPC   | Accuracy  |  0.8319  |  0.8336  |  0.8371  |
> |                         |    CoLA   | MCC       |  0.5636  |  0.5636  |  0.5756  |
> |                         | SQuAD 2.0 | F1-score  |  78.8113 |  78.7647 |  78.9547 |
> | **Stable Diffusion v2** |    COCO   | FID-score |  27.2551 |     -    |  25.6262 |

---

> ### Author Response · Authors · 2023-11-23
> **Thanks for your valuable feedback! (cont.)**
>
> **Your Questions**
> > - The paper seems to over-sell itself in many places and that space could be used to add more detail.
> > - It would be clearer to show the bitwidth in the formats in the tables since these comparisons are across 32, 16, and 8 bits.
> > - What FP8 variant is used for comparison? E4M3 typically has the highest accuracy.
> > - Equation 1 can could be explained better in the text. For example, are the X elements an arbitrary tensor? It is unclear what the purpose of this equation is overall.
> > - Equations 2-5 could be aligned to make them more readable.
>
> **Q6.** **Comments on Paper Details.**
>
> **A6.** We have incorporated your comments into the revised paper as follows:
>
> (a) You requested detailed configurations of various formats used in the experiments, including FP8. Accordingly, we have added detailed descriptions of MSFP, BSFP, and FP8 in Section 4, Tables 2 and 3. The specifics are as follows:
> - MSFP: 1-bit sign, 8-bit exponent, and 7-bit mantissa.
> - BSFP: 5-bit and 2-bit mantissa for two subwords, and 8-bit and 7-bit  two scale factors.
> - FP8: 1-bit sign, 4-bit exponent, and 3-bit mantissa (E4M3).
>
> (b) In response to your suggestion, we have removed Equation 1 and revised Equations 2-5 for enhanced readability.
>
> (c) We have addressed the issue of overselling in the initial draft and added additional details for clarity. For example, in Section 3, we have included further explanations and details about representation, making the paper more comprehensible to our readers.
>
> --------
>
> **Your Question**
> > What is the definition for exponent baseline? Initially I thought the exponent would be added to these but the baseline for mode 2 includes the exponent bits themselves.
>
> **Q7. Definition of exponent baseline**
>
> **A7.** Your understanding of the exponent baseline is accurate. To clarify the definition of the exponent baseline once more: "The exponent baseline sets a default value for the exponent in each representation." We have identified and corrected a typographical error regarding the exponent baseline in the original representation 2. The revised exponent baseline in representation 2 is -2. We deeply appreciate your meticulous observation and feedback.
>
> ---------
>
> **Your Question**
> > The ALU seems like it is purely doing standard floating-point multiplication? Or does it support additional functionality to justify its name?
>
> **Q8. SuFP ALU**
>
> **A8.** We would like to explain SuFP ALU. The SuFP ALU is designed to perform multiply-accumulate operations on the mantissa and exponent of SuFP. Specifically, it includes the capability to multiply two mantissa and add exponent. Generally, an ALU (Arithmetic Logic Unit) is defined as a unit that performs arithmetic operations and various bit-level logic operations. Therefore, the name SuFP ALU was chosen to avoid confusion and to emphasize its specialized functionality in handling SuFP operations.
>
> ---------
>
> **Your Question**
> > Are there any additional floating-point quantization scale factors with this method? FP8 method often still use additional higher-precision quantization scales still.
>
> **Q9.** **High precision scale factor for floating-point quantization**
>
> **A9.** SuFP adopts multi-region piecewise quantization and tensor-wise scalable bias to represent data distribution effectively. One of the primary advantages of this approach is that it does not require additional high-precision quantization scale factors. This structure reduces the hardware overhead required for scale factor processing, allowing for more compact processing elements to be implemented. This provides significant benefits, especially in terms of hardware efficiency and performance optimization.
>
> -------
> We sincerely appreciate the time and effort you dedicated to reviewing our paper. Your insightful feedback has significantly contributed to enhancing the quality of our work. We have carefully considered and addressed your concerns in our revisions, and we hope that these changes meet your expectations. We value your input immensely and remain open to any further discussions or suggestions that could further enrich our research. Your continued guidance and questions are greatly welcomed, and we look forward to any further suggestions or questions you may have.

---

### Official Review · Reviewer_CpEu · 2023-11-06

**Soundness:** 3 good
**Presentation:** 3 good
**Contribution:** 3 good
**Rating:** 6
**Confidence:** 5

**Summary:**

To solve the huge dynamic ranges for outliers problems in quantization, this paper introduces a new data type and corresponding quantization method to improve both memory footprint and logic efficiency. The key idea of SuFP is multi-region piecewise
quantization using a tensor-wise scalable bias which offers flexible adaptability to diverse data distributions. Furthermore, the tailored
hardware for SuFP is also provided which employs only integer arithmetic units and shifters. The evaluation has been processed in different tasks, such as vision, language, and generative models.

**Strengths:**

1. Good writing style. The paper is easy to follow.
2. The paper focuses on a great problem of quantization "outlier" which is critical to the quantization accuracy.
3. Multiple tasks are included in the experiments, which proves the method's flexibility.
4. The works incorporate both the algorithm with the hardware into consideration.

**Weaknesses:**

1. The paper mainly compares different quantization schemes but does not incorporate different quantization frameworks in the experiment comparisons.
2. The hardware setup details are not clear.
3. The hardware efficiency evaluation only provides a normalized result without a specific number, which may cause additional difficulty for future works' comparison.

**Questions:**

Please refer to the weakness.

---

> ### Author Response · Authors · 2023-11-23
> **Thanks for your valuable feedback!**
>
> Thank you for your detailed feedback. In response to the points you raised, we have supplemented our paper with additional data. We gratefully acknowledge the significant role your valuable insights have played in refining our paper.
>
> **Q1. The paper mainly compares different quantization schemes but does not incorporate different quantization frameworks in the experiment comparisons.**
>
> **A1.** Thank you for your valuable feedback. In response to your suggestion, we have extended the evaluation section of our paper. This extension includes additional experiments with the BERT model and Stable Diffusion v2, utilizing advanced techniques like MSFP and BSFP. We carefully selected the most effective configurations for MSFP and BSFP to ensure the experiments showcase optimal performance.
> The results of these additional experiments are as follows:
>
> The table below shows that SuFP performs better than all other methods in BERT and Stable Diffusion v2.
>
> | **Model**               | **Benchmark** | **Metric** | **MSFP** | **BSFP** | **SuFP** |
> |-------------------------|:---------|:-----------|:--------:|:--------:|:--------:|
> | **BERT-base**           |    MRPC   | Accuracy  |  0.8319  |  0.8336  |  0.8371  |
> |                         |    CoLA   | MCC       |  0.5636  |  0.5636  |  0.5756  |
> |                         | SQuAD 2.0 | F1-score  |  78.8113 |  78.7647 |  78.9547 |
> | **Stable Diffusion v2** |    COCO   | FID-score |  27.2551 |     -    |  25.6262 |
>
> **Q2, Q3. The hardware setup details are not clear. The hardware efficiency evaluation only provides a normalized result without a specific number, which may cause additional difficulty for future works' comparison.**
>
> **A2.** I appreciate your insightful suggestions on our hardware setup and efficiency evaluation. We recognize the importance of detailed and clear information to enhance the credibility and comparability of our research. In response, we have added extensive details about our hardware implementation, including the setup for each Processing Element (PE), in Appendix F. Additionally, we now provide specific numbers accompanied by hardware details, clearly summarized in the table below for ease of reference and comparison.
>
> |                | **16x FP32** | **16x BF16** | **16x FP8** | **16x BSFP** | **MSFP**  | **SuFP**  |
> |:--------------|:------------:|:------------:|:-----------:|:--------------:|:-----------:|:-----------:|
> | **Area (um2)** |  29731.9679  |   7430.9759  |  5205.3120  | 23829.1200   | 3485.7900 | 3303.7200 |
> | **Power (mW)** |    19.8528   |    3.1024    |    1.5173   | 9.6144       | 1.2051    | 1.1649    |
>
> --------
>
> We sincerely hope that these revisions effectively address your concerns. Your insightful feedback has been invaluable in enhancing the quality of our work, and we are deeply grateful for it. We remain open to further discussions and are eager to engage in any additional dialogue that may benefit our research. Your input is greatly appreciated, and we look forward to any further suggestions or questions you may have.

---

### Official Review · Reviewer_tqWQ · 2023-11-07

**Soundness:** 2 fair
**Presentation:** 2 fair
**Contribution:** 2 fair
**Rating:** 3
**Confidence:** 4

**Summary:**

This paper proposes a new data type, Super Floating-Point (SuFP), to improve both memory footprint and computational efficiency for deep neural network quantization. SuFP utilizes multi-region piecewise quantization with tensor-wise scalable bias, allowing for optimized precision for different data regions and adaptability to various data distributions. Experiments show that compared to FP8, SuFP achieves 1.58x and 1.30x improvement in computational capability and energy efficiency, respectively, without losing model accuracy performance.

**Strengths:**

+ The paper is well-written and easy to follow. Illustrative figures are well plotted and easy to understand.
+ Co-designing the hardware MAC architecture for the proposed data type SuFP enables better hardware efficiency.
+ The evaluation benchmarks are diverse, including both vision and language tasks.

**Weaknesses:**

+ The ablation study lacks the improvement breakdown on piecewise data representation and tensor-wise scalable bias.
+ The accuracy evaluation experiments lack results on MSFP and BSFP for larger models (Table 3). The efficacy of the proposed SuFP on Large Language Models such as LLaMa2 is also unclear.
+ The proposed SuFP saves 6% memory, with 1.05x throughput improvement and 1.03x energy savings over MSFP (MX9). The improvement is marginal.
+ MSFP have multiple versions: MX4, MX6 and MX9. It is unclear how SuFP narrows its bit width.

**Questions:**

Please answer the questions in the weakness section.

---

> ### Author Response · Authors · 2023-11-23
> **Thanks for your valuable feedback!**
>
> Thank you for your detailed feedback. We have carefully reviewed the points you raised and, accordingly, conducted additional experiments and analyses to enhance the depth and quality of the paper. We would like to acknowledge that your valuable insights have significantly contributed to improving the quality of the paper.
>
> **Q1. The ablation study lacks the improvement breakdown on piecewise data representation and tensor-wise scalable bias.**
>
> **A1.** Thank you for providing your feedback. To provide evidence of the improvements in SuFP and to reinforce our arguments, we have conducted additional experiments.
>
> Firstly, we have carried out a detailed analysis of piecewise data representation. We compared the performance of our method with existing piecewise quantization methods, and the results of this have been added to Appendix B. Through these experiments, we have verified the effectiveness of our method in efficiently representing both dense regions around zero and wide-ranging sparse regions. Unlike traditional piecewise quantization, which lacks flexibility in bit allocation across different areas, our method applies multi-region piecewise quantization, enabling optimized bit allocation for each area through variable encoding fields and data fields.
>
> Furthermore, we conducted additional experiments on tensor-wise scalable bias. By analyzing tensor-wise and channel-wise data distributions, we confirmed that the tensor distribution significantly reflects the channel distribution. This finding has been incorporated into Appendix C. Based on this, we have demonstrated the effectiveness of tensor-level quantization and proved that our quantization method using tensor-wise scalable bias accurately represents tensor data distributions.
>
> Lastly, we have confirmed that our method exhibits strong robustness against dependencies on outliers, as detailed in Appendix D. Considering the impact of outliers on accuracy, this robustness is considered a significant advantage of our data type.
>
> **Q2. The accuracy evaluation experiments lack results on MSFP and BSFP for larger models (Table 3). The efficacy of the proposed SuFP on Large Language Models such as LLaMa2 is also unclear.**
>
> **A2.** Thank you for your suggestions. We have actively incorporated your feedback by implementing MSFP and BSFP in detail, and conducted additional experiments for a direct performance comparison with our method. The results obtained from this process have been included in the paper to enhance its completeness. These comparative performance outcomes can be found in Section 4. As shown in the table below, located in Table 3 of the same section, illustrates that SuFP exhibits consistent performance in various language and text-to-image models.
>
> | **Model**               | **Benchmark** | **Metric** | **MSFP** | **BSFP** | **SuFP** |
> |:-------------------------|:---------|:-----------|:--------:|:--------:|:--------:|
> | **BERT-base**           |    MRPC   | Accuracy  |  0.8319  |  0.8336  |  0.8371  |
> |                         |    CoLA   | MCC       |  0.5636  |  0.5636  |  0.5756  |
> |                         | SQuAD 2.0 | F1-score  |  78.8113 |  78.7647 |  78.9547 |
> | **Stable Diffusion v2** |    COCO   | FID-score |  27.2551 |     -    |  25.6262 |
>
> Furthermore, following your suggestion, we have conducted additional experiment on Large Language Models, the Llama 2. The results showed a difference of approximately 6% compared to FP16. This suggests the potential applicability of SuFP to LLMs, and we have also identified further opportunities for optimization in this regard.
>
> |**Model**| **Benchmark**|**FP16** | **FP8** | **MSFP** | **SuFP** |
> |:-----------|:---------------|:--------:|:-------:|:----------:|:----------:|
> | **Llama 2** |       STEM      |   0.369  |  0.214  |   0.372  |  0.349   |
> |             |    Humanities   |   0.433  |  0.242  |   0.431  |  0.403   |
> |             | Social Sciences |   0.518  |  0.217  |   0.523  |  0.486   |
> |             |      Other      |   0.525  |  0.238  |   0.524  |  0.493   |
> |             |     Average     |   0.459  |  0.229  |   0.460  |  0.430   |

---

> ### Author Response · Authors · 2023-11-23
> **Thanks for your valuable feedback! (cont.)**
>
> **Q3.** **The proposed SuFP saves 6% memory, with 1.05x throughput improvement and 1.03x energy savings over MSFP (MX9). The improvement is marginal.**
>
> **A3.** We appreciate your comment. I would like to provide additional perspective on the issue of marginal performance improvement you have pointed out. As you rightly noted, the performance enhancement at the processing element (PE) level may seem marginal. However, this difference signifies an important improvement from the standpoint of hardware design and actual implementation. Particularly, from the perspective of scalability, which involves using multiple PEs for parallel operations, our method exhibits notable advantages.
>
> Block floating-point data types like MSFP and BSFP utilize a shared exponent or scale factors at the block level, requiring an increase in the number of PEs to enhance scalability. Furthermore, considering scalability and increasing the number of inputs results in larger block sizes, which in turn leads to a reduction in accuracy. In contrast, our proposed method of using tensor-wise bias allows for scaling by simply increasing the number of inputs to each PE. This consideration is vital in hardware implementation.
>
> In actual hardware design, this difference translates into a substantial benefit. Increasing the number of PEs implies a rise in the necessary registers, multiplexers, accumulators, control logic, and network interfaces, leading to considerable hardware resource overhead. Conversely, adjusting the number of inputs processed by each PE to enable parallel operations is more advantageous in reducing hardware overhead and design complexity, which are key elements of efficient system design. Therefore, although the gain at the PE level might seem minor, from the perspective of overall system design and practical implementation, our proposed approach offers noteworthy advantages.
>
> **Q4.** **MSFP have multiple versions: MX4, MX6 and MX9. It is unclear how SuFP narrows its bit width.**
>
> **A4.** As you rightly pointed out, reducing the bit-width is indeed a crucial consideration in quantization and hardware design. However, considering that hardware, once manufactured, is challenging to modify, we focused on selecting an optimal data bit-width configuration during the initial design phase to ensure peak performance.
>
> This decision was made from a hardware optimization standpoint. By employing a combination of variable encoding and data fields, we were able to finely balance granularity and dynamic range. This approach allowed us to optimize model performance and efficiency within the fixed constraints of the hardware's bit-width.
>
> --------
> Based on these experimental results and analyses, we have an improvement breakdown of our proposed method in the paper. Once again, We sincerely appreciate your insightful and meticulous review, which has significantly contributed to the advancement of our paper. Your comments and suggestions have played a vital role in enriching the logical structure and content of our paper, thereby enhancing the quality of our research. We are truly grateful for providing us with this opportunity to develop our study further.

---

### Meta-Review · Area_Chair_Yu4V · 2023-12-11

**Metareview:**

This paper proposes a new floating point data type, with improved memory footprint and computational efficiency for neural network inference (post training quantization, not training), and shows some favorable experimental comparisons relative to FP8.

Concerns remained that compared to other existing floating point formats based on similar block-wise representations, the gains might not yet be substantial enough to warrant acceptance at ICLR (and that experiments would need to be further expanded), also given that some of the previous formats addressed the harder task of training as well, whereas here only inference is demonstrated.

We hope the detailed feedback helps to strengthen the paper for a future occasion.

**Justification For Why Not Higher Score:**

Similarity to existing block-wise FP formats and respective gains achieved

**Justification For Why Not Lower Score:**

N/A

---

### Decision · Program_Chairs · 2024-01-16

Reject